# Synthesis of Si-Modified Pseudo-Boehmite@kaolin Composite and Its Application as a Novel Matrix Material for FCC Catalyst

**DOI:** 10.3390/ma15062169

**Published:** 2022-03-15

**Authors:** Chengyuan Yuan, Zhongfu Li, Lei Zhou, Guannan Ju

**Affiliations:** School of Materials Science and Engineering, Shandong University of Technology, Zibo 255000, China; yuanchengyuan_sdut@163.com (C.Y.); lzf1632008@126.com (Z.L.); yshxq12138@163.com (L.Z.)

**Keywords:** kaolin, Si modification, pseudo-boehmite, matrix material, composite, FCC catalyst, catalytic cracking

## Abstract

Fluid catalytic cracking (FCC) has been the primary processing technology for heavy oil. Due to the inferior properties of heavy oil, an excellent performance is demanded of FCC catalysts. In this work, based on the acid extracting method, Si-modified pseudo-boehmite units (Si-PB) are constructed in situ and introduced into the structure of kaolin to synthesize a Si-PB@kaolin composite. The synthesized Si-PB@kaolin is further characterized and used as a matrix material for the FCC catalyst. The results indicate that, compared with a conventional kaolin matrix, a Si-PB@kaolin composite could significantly improve the heavy oil catalytic cracking performance of the prepared FCC catalyst because of its excellent properties, such as a larger surface area, a higher pore volume, and a good surface acidity. For the fresh FCC catalysts, compared with the FCC catalysts using conventional kaolin (Cat-1), the gasoline yield and total liquid yield of the catalyst containing Si-PB@kaolin (Cat-2) could obviously increase by 2.06% and 1.55%, respectively, with the bottom yield decreasing by 2.64%. After vanadium and nickel contamination, compared with Cat-1, the gasoline yield and total liquid yield of Cat-2 could increase by 1.97% and 1.24%, respectively, with the bottom yield decreasing by 1.80 percentage points.

## 1. Introduction

Fluid catalytic cracking (FCC) is an important process for crude oil, in which FCC catalysts have played a key role [1,2,3]. Recently, with crude oil becoming increasingly inferior, the demand for an excellent catalytic cracking performance of FCC catalysts has greatly increased [4,5,6,7,8]. Usually, FCC catalysts are made of zeolites, a binder, and a matrix material. For heavy oil catalytic cracking, an FCC catalyst not only needs to be of high quality for zeolites, but also needs to exhibit an excellent performance for the matrix material, with qualities such as a large surface area, a high pore volume and abundant surface acidity to promote the diffusion and pre-cracking of heavy oil molecules in the FCC catalyst [9,10,11,12,13].

Kaolin has been mostly used as the matrix material for FCC catalyst preparation because of its attractive physico-chemical properties, such as excellent hydro-thermal stability, high mechanical strength, and good thermal conductivity [14,15]. However, conventional kaolin commonly lacks pore structure and surface acid sites, which has made the conventional kaolin unsatisfactory for the present preparation demands of FCC catalysts [16,17,18]. Therefore, increasing attention has been paid to the synthesis of various modified kaolin materials with large surface areas, high pore volumes and good surface acidity [19]. Among them, acid extraction is the most-used method to improve the surface area, pore volume and surface acidity for conventional kaolin by extracting alumina species from the framework of kaolin [20,21]. However, the method of acid extraction has been seriously restricted in terms of practical application because of issues such as material loss and acidic waste solution [22]. Pseudo-boehmite (PB) is a kind of aluminum-based matrix material for FCC catalysts with the formula of AlOOH·nH_2_O (n = 0.08 − 0.62) and possesses a large surface area, a high pore volume and plenty of acid sites in contrast with kaolin [23]. Recently, it has been reported that Si modification could improve the surface area, the pore volume and the surface acidity of conventional pseudo-boehmite even further [24,25]. 

Based on the above fact, to improve the structural properties and surface acidity of conventional kaolin, Si-modified pseudo-boehmite (Si-PB) was constructed in situ and introduced into the structure of kaolin based on the acid-extraction method to synthesize the Si-modified pseudo-boehmite@kaolin composite material (Si-PB@kaolin) described in this work. Compared with a conventional kaolin matrix material, the surface area, pore volume, surface acidity and cracking activity of Si-PB@kaolin are all much higher. As a matrix material, the synthesized Si-PB@kaolin could greatly improve the heavy oil catalytic cracking performance of prepared FCC catalysts. More importantly, the synthesis method of Si-PB@kaolin in this work could also avoid the above-mentioned issues of acid extraction, which gives it good practical application prospects.

## 2. Materials and Methods

### 2.1. Materials

Industrial-grade Kaolin, REY zeolite and alumina sol were provided by Shanxi Tengmao Technology Company, Hejin, China. Analytically pure Na_2_SiO_3_·9H_2_O, NaAlO_2_, and H_2_SO_4_ were purchased from Sinopharm Chemical Reagent Company, Shanghai, China. All chemicals were used as received without further purification.

### 2.2. Synthesis of Si-PB@kaolin Composite

The synthesis procedure of the Si-PB@kaolin composite is shown in Figure 1. As shown, kaolin was first calcined at 850 °C for 2 h. Then, 50 g of calcined kaolin was mixed with 250 mL of H_2_SO_4_ solution and stirred at 90 °C for 3 h to carry out acid extraction. After this, 50 mL aqueous solutions of 10 g Na_2_SiO_3_·9H_2_O and 50 mL aqueous solution of 4 g NaAlO_2_ were simultaneously added into the above slurry dropwise. Then, the pH of the above-obtained slurry was adjusted to around 8.5 by ammonia water, in which the secondary units of Si-PB were constructed in the structure of kaolin through a neutralization reaction with the extracting Al species [26]. After this, the obtained slurry was statically aged for another 4 h and then filtrated, washed and dried to obtain the Si-PB@kaolin composite. Before the FCC catalyst preparation, the synthesized Si-PB@kaolin composite was necessarily ion exchanged by NH_4_Cl solution to decrease its Na_2_O contents to an appropriate range. 

### 2.3. Preparation of FCC Catalysts 

Compared FCC catalyst: REY zeolite, alumina sol, conventional kaolin and deionized water were fully mixed, and then the above slurry was spray-dried to make a micro-sphere FCC catalyst (Cat-1). The components’ mass ratio for Cat-1 was REY: alumina sol (based on Al_2_O_3_): conventional kaolin = 35:10:55.

FCC catalyst containing Si-PB@kaolin: First, REY zeolite, alumina sol, Si-PB@kaolin, conventional kaolin and deionized water were fully mixed, and then the above slurry was spray-dried to make a micro-sphere FCC catalyst (Cat-2). The components’ mass ratio for Cat-2 was REY: alumina sol (based on Al_2_O_3_): Si-PB@kaolin: conventional kaolin = 35:10:10:45.

### 2.4. Characterizations and Evaluations

X-ray diffraction (XRD, Rigaku corporation, Akishima-shi, Tokyo, Japan) was carried out on a Rigaku D/max-2200 PC X-Ray diffractometer with CuK-Alpha radiation (*k* = 0.15418 nm), operating at 40 kV, 40 mA, and scanning from 5° to 75° at a speed of 0.01 °/s. N_2_ adsorption–desorption measurement at −196 °C was performed on a Micromeritics ASAP 2010 instrument (Micromeritics instrument corporation, Norcross, GA, USA) to characterize the textural properties. The Brunauer–Emmett–Teller (BET) method and the Barrett–Joyner–Halenda (BJH) method were used to determine the surface areas and pore volumes of the samples. NH_3_ temperature programmed desorption (NH_3_-TPD) was performed on a Micromeritics AUTOCHEM II 2920 (Micromeritics instrument corporation, Norcross, GA, USA) in the range of 100–500 °C at a heating rate of 15 °C/min. The adsorption of ammonia on the samples was performed at room temperature, followed by removing physically adsorbed ammonia at 100 °C for 1 h in pure flowing nitrogen. The Fourier transform infrared spectra after pyridine adsorption (Py-FTIR) and 2,6-di-tert-butylpyridine (DTBPy-FTIR) were obtained on a Bruker Tensor 27 FT-IR spectrometer (Bruker Company, Karlsruhe, Germany). All samples were activated at 300 °C for 3 h before Py and DTBPy adsorption. Physically adsorbed Py and DTBPy on acid sites were removed by a vacuum molecular pump at 150 °C for 2 h. The micro activity test (MAT) that was used to evaluate the cracking activities of FCC catalysts treated by hydro-thermal processing was performed on micro-cracking reaction equipment (WF-2006, Beijing Huayang, Beijing, China). The scanning electron microscopy (SEM, Hitachi Ltd., Tokyo, Japan) micrographs were obtained by a Hitachi S4800 electron microscope with 1 nm resolution at 15 kV and magnification from 30× to 800,000×.

The heavy oil catalytic cracking evaluations for FCC catalysts treated by hydro-thermal processing were performed at 530 °C on an advance cracking evaluation (ACE) unit developed by Kayser Technology Inc. Company, Houston, Texas, America, with the catalyst/oil weight ratio being 5. Gaseous products were analyzed using a GC-3000 online chromatograph produced by INFICON Company (New York, NY, USA) according to the UOP method 539. GC-3000 used four chromatographic modules for detection, and the work temperature was 0–50 °C. The carrier gases used were helium, hydrogen, nitrogen, and argon. Simulated distillation of liquid products was carried out using a 7890B chromatograph produced by Agilent Technologies Inc. (Santa Clara, CA, USA) according to the SH/T 0558 procedure. The working environmental temperature of Agilent 7890B was 15–35 °C. Retention time repeatability was <0.0008 min. Carrier and makeup gas settings were selectable for helium, hydrogen, nitrogen, and argon/methane. Coke deposited on the catalyst was quantified with a CO_2_ analyzer produced by Servomex Group Co., Ltd. (Sussex, England, UK). The detection range of the CO_2_ analyzer was 0–20%. When the detection value was lower than 0.4%, the regeneration step was considered to be completed. The conversion and yields of dry gas (H_2_ + C_1_ + C_2_), LPG (liquefied petroleum gas) (C_3_ + C_4_), gasoline (C_5_ < bp < 221 °C), diesel (221 °C < bp < 343 °C), bottom (bp > 343 °C), and coke were calculated. The paraffin (P), olefin (O), naphthene (N), and aromatic (A) contents (PONA analysis) of the cracked gasoline were analyzed by gas chromatography (Varian CP-3380). The heavy oil was provided by Lanzhou Petrochemical Company, Lanzhou, China. The properties of heavy oil are listed in Table 1.

## 3. Results

### 3.1. Characterization Results 

The XRD patterns for kaolin, calcined kaolin and the Si-PB@kaolin composite are exhibited in Figure 1. As shown, the characteristic diffraction peaks for conventional kaolin appeared at around 12.3°, 20.2°, 21.3°, 24.8°, 35.0°, 36.0°, 38.4°, 39.2°, 45.5°, 55.2°, and 62.5°, respectively, which are in good agreement with the standard XRD pattern of kaolinite [27]. For the calcined kaolin and Si-PB@kaolin composite samples, the characteristic diffraction peaks of kaolin disappeared, leaving a wide diffraction peak in the range of 15–30° due to the structure destruction caused by calcination [28]. Compared with calcined kaolin, several new wide diffraction peaks appeared at around 13.5°, 38.0°, 49.3° and 65.4°, respectively, for the Si-PB@kaolin composite, which could be attributed to the characteristic diffractions of Si-PB and suggested the successful construction of Si-PB secondary units into the structure of kaolin [29].

Figure 2 shows the N_2_ adsorption–desorption isotherms for the conventional kaolin, the calcined kaolin and the Si-PB@kaolin composite. As shown, both the conventional kaolin and the calcined kaolin displayed the characteristic N_2_ adsorption–desorption isotherms of macro-porous materials, indicating that the conventional kaolin and calcined kaolin mainly contained macro-pores that were derived from the accumulation of particles [30]. Compared with the conventional kaolin and calcined kaolin samples, the N_2_ adsorption–desorption isotherm for the Si-PB@kaolin composite was very different and displayed the isotherm feature of meso-porous material with an obvious hysteresis loop in the range of P/P_0_ = 0.4–1, which indicated that there were plenty of meso-pores in the structure of the Si-PB@kaolin composite due to the construction of Si-PB secondary units [31].

The pore structure properties for the conventional kaolin, the calcined kaolin and the Si-PB@kaolin composite are listed in Table 2. It can be seen that the surface areas and pore volumes for the conventional kaolin were 23 m^2^/g and 0.10 cm^3^/g, respectively. After calcination, the surface areas and pore volumes for the calcined kaolin decreased to 9 m^2^/g and 0.04 cm^3^/g, respectively, due to the structure destruction. In contrast, because of the introduction of the Si-PB secondary units, the surface areas and pore volumes for the Si-PB@kaolin composite clearly increased to 106 m^2^/g and 0.28 cm^3^/g, respectively, which were much higher values than those of the conventional kaolin and the calcined kaolin. 

Surface acidity is an important property of matrix materials for FCC catalysts. The NH_3_-TPD profiles for the conventional kaolin, the calcined kaolin and the Si-PB@kaolin composite are exhibited in Figure 3. As shown, the conventional kaolin, the calcined kaolin and the Si-PB@kaolin composite all displayed a broad NH_3_-desorption peak in the temperature range of 100–450 °C. Compared with the conventional kaolin and the calcined kaolin, the NH_3_-desorption peak area for the Si-PB@kaolin composite was much higher, indicating that there were much more surface acid sites on the Si-PB@kaolin composite [32].

Py-FTIR was also used to confirm the type of acid sites for the conventional kaolin, the calcined kaolin and the Si-PB@kaolin composite. As shown in Figure 4, the conventional kaolin and the calcined kaolin only exhibited an IR band at around 1450 cm^−1^ that was attributed to L acid sites [33], which suggested that the surface acid sites for the conventional kaolin and the calcined kaolin were mainly L acid sites. However, compared with the conventional kaolin and the calcined kaolin, in addition to the band at around 1450 cm^−1^, the Si-PB@kaolin composite exhibited a new IR band at around 1540 cm^−1^ that could be attributed to B acid sites [34], indicating that both L acid sites and B acid sites existed on the surface of the Si-PB@kaolin composite. It has been widely accepted that B acid sites are very favorable for catalytic cracking and are usually non-existent in conventional matrix materials such as kaolin and pseudo-boehmite for FCC catalysts [35], which would make the Si-PB@kaolin composite an ideal matrix material for FCC catalysts. 

The specific acid site quantities for the conventional kaolin, the calcined kaolin and the Si-PB@kaolin composite are listed in Table 2. As shown, the total quantity of acid sites of the Si-PB@kaolin composite was significantly higher than that of the conventional kaolin and the calcined kaolin and could reach to 97.7 μmol/g, which could be attributed to the introduction of Si-PB secondary units. Moreover, in addition to L acid sites, there were also 11.3 μmol/g B acid sites for the Si-PB@kaolin composite.

### 3.2. Characterizations for Prepared FCC Catalysts 

The physico-chemical properties for prepared FCC catalysts are listed in Table 3. As shown, the surface areas and pore volumes for Cat-1 with the conventional kaolin as the matrix material were 201.2 m^2^/g and 0.25 cm^3^/g, respectively. Compared with Cat-1, the surface areas and pore volumes for Cat-2 containing the Si-PB@kaolin composite clearly increased to 227.5 m^2^/g and 0.33 cm^3^/g, respectively. Furthermore, because of the excellent properties of the Si-PB@kaolin composite, such as a large surface area, a high pore volume and abundant acid sites, the S_meso_, V_meso_, total acid sites, B/L and MAT for Cat-2 all obviously increased in contrast with Cat-1.

In addition to the above physico-chemical properties, the acid site accessibility has also been considered as an important parameter for FCC catalysts [36]. Herein, Py-FTIR and DTBPy-FTIR were used to estimate the acid sites accessibilities for the prepared FCC catalysts through the method reported by Yuan et al. [37]. Figure 5 shows the Py-FTIR and DTBPy-FTIR values for the prepared FCC catalysts. As shown, the characteristic adsorption peaks for Py and DTBPy appeared at around 1540 cm^−1^ and 3400 cm^−1^, respectively.

The characteristic peak area ratios (A_Py_/A_DTBPy_) derived from FTIR for the prepared FCC catalysts are shown in Figure 6. As reported by Yuan et al., for different FCC catalysts, a higher value of A_Py_/A_DTBPy_ means higher accessibility to acid sites [37]. It can be seen from Figure 6 that the A_Py_/A_DTBPy_ value of Cat-2 was much bigger than that of Cat-1, suggesting that Si-PB@kaolin could obviously enhance the accessibility to acid sites of a prepared FCC catalyst compared with conventional kaolin matrix material.

Figure 7 shows the SEM images for the prepared FCC catalysts. It can be seen from Figure 7a,b that both the micro-spheres of Cat-1 and Cat-2 possessed good sphericity. Compared with Cat-1 using the conventional kaolin matrix material (Figure 7c), the micro-sphere surface of Cat-2 containing the Si-PB@kaolin composite matrix material (Figure 7d) seemed to be much less compacted. Further, it can be clearly seen from the images of the surface detail that there were much more gaps and pores on the surface of Cat-2 (Figure 7f) in contrast with Cat-1 (Figure 7e), which shows that Cat-2 would be greatly helpful for the diffusion of heavy oil molecules and thus could promote the use of Cat-2 for heavy oil catalytic cracking.

### 3.3. Evaluation Results for Heavy Oil Catalytic Cracking 

The heavy oil catalytic cracking performances of the fresh FCC catalysts are listed in Figure 8. It can be seen that the heavy oil catalytic cracking performance of Cat-2 obviously improved by using the Si-PB@kaolin composite as matrix in contrast with Cat-1. Compared with Cat-1, the bottom yield of Cat-2 decreased by 2.64%; meanwhile, the gasoline yield and total liquid yield (LPG+gasoline+diesel) of Cat-2 were obviously increased by 2.06% and 1.55%, respectively, suggesting the excellent performance of heavy oil catalytic cracking for Cat-2.

The prepared FCC catalysts were also contaminated by V and Ni through the impregnation method (V: 5000 ppm; Ni: 3000 ppm) to investigate the differences in their anti-heavy metal contamination performances. As shown in Figure 9, Cat-2 exhibited a better anti-heavy metal contamination performance than Cat-1. Compared with Cat-1, the bottom yield of Cat-2 decreased by 1.80% with gasoline yield and total liquid yield increasing by 1.97% and 1.24%, respectively, which indicated the good anti-heavy metal contamination performance of Cat-2, containing the Si-PB@kaolin composite matrix material.

The results of the PONA analysis and the research octane number (RON) of cracked gasoline are shown in Table 4. As shown, compared with Cat-1, the paraffin content of gasoline for Cat-2 was lower, while the amounts of olefins, naphthenes and aromatics were higher, which resulted in a higher RON for Cat-2.

## 4. Conclusions

In this work, a Si-modified pseudo-boehmite@kaolin composite (Si-PB@kaolin) was synthesized by an in situ construction method based on acid extraction and further used as a novel matrix material for FCC catalyst preparation. In contrast with conventional kaolin matrix material, the properties of the Si-PB@kaolin composite, such as a large surface area, pore volume and surface acidity, were significantly promoted by the introduction of Si-modified pseudo-boehmite secondary units, which created excellent physico-chemical properties and acid site accessibility for the prepared FCC catalyst. The evaluation results suggested that, compared with the catalyst using conventional kaolin matrix material, the FCC catalyst containing Si-PB@kaolin exhibited much better heavy oil catalytic cracking and anti-heavy metal contamination performances. Furthermore, the synthesis procedure of the Si-PB@kaolin composite is simple and low-cost, which gives it good prospects for application. 

## Data Availability

Not applicable.

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
