# Peer review of "Synthesis of Si-Modified Pseudo-Boehmite@kaolin Composite and Its Application as a Novel Matrix Material for FCC Catalyst"

_materials, 2022, doi:10.3390/ma15062169_

Round 1
Reviewer 1 Report
The manuscript is interesting, especially the modification of kaolin by Si exchange in its structure and the application of the FCC reaction, due to the multiple diffusion and reaction factors that these generate in interaction with catalysts. However, it is necessary to consult a greater amount of literature, more updated, as the most recent is from 2019, two years from the current date. There is a lot of literature related to FCC reactions that could explain some of the phenomena observed in his study.
It is necessary to refine the XRD results to reduce the noise because it is confused with the crystallinity that is mentioned in the angles of 13.5o, 38.0o, 49.3o and 65.4o.
Comments:
- In section 2.2. Synthesis of the Si-PB@kaolin compound; line 69 mentions: After that, the Na2SiO3 and NaAlO2 solutions were added simultaneously into the above suspension until the pH of the system approached about 8.5, however, it is not specified at what relation weight or mol, the Na2SiO3, and NaAlO2 were added to kaolin. And how the pH of 8.5 is reached or what reagent is added to reach it? You need to complete this section with more information to clarify.
- In line 99 it is mentioned that the catalyst/oil catalytic cracking weight ratio being 5, that is, for each portion of catalyst, 1/5 of oil was added? How is this relationship explained or why was this relationship used?
-
Line 113, mentioned: For calcinated kaolin and Si-PB@kaolin composite samples, the characteristic diffraction peaks of kaolin disappeared leaving a wide diffraction peak in the range of 15-30 due to the destruction structure caused by calcination. But what happens during the calcination process until 800 °C, How is it transformed into semi amorphous material?
- Line 118 mentions that several new wide diffraction peaks appeared at around 13.5 o, 38.0 o, 49.3 o and 65.4 o respectively for Si-PB@kaolin composite, which could be attributed to the characteristic diffractions of Si-PB and suggested the successful construction of Si-PB secondary units into the structure of kaolin. But what will be the crystalline system that it presents? will it change from monoclinic to some other system or network by structural reorganization?
-
Line 138 mentions that: On the contrary, due to the introduction of secondary Si-PB units, the surface area and pore volume for the Si-PB@kaolin compound obviously increased to 106 m2/g and 0.28 cm3 /g respectively, which were much higher than that of conventional kaolin and calcined kaolin. But do you mean that these secondary units are the ones that contribute to the surface area? How to know if in DRX it was commented that these are identified as independent structures? Can these units be found inside kaolin as illustrated in Scheme 1? can you comment something about this?
-
Figure 4. Py-FTIR spectra for kaolin, AT-kaolin and Si-PB@kaolin, the signal between 1400 and 1500 cm-1, to which species does it correspond or what could it indicate?
-
Line 184 indicates that the value of Cat-2 was much bigger than that of Cat-1, suggesting that Si-PB@kaolin could obviously enhance the acid sites accessibility of prepared FCC catalyst in compariso with conventional kaolin matrix material. Can you explain, why is it acidity changed? what bonds in its union will be present or are the SiPB blocks that cause said the increase, and how are they organized or structurally united?
-
From the Ca2 sample. It would be interesting to show a TEM to observe its structural organization.
-
The prepared FCC catalysts were also contaminated by V (5000 ppm) and Ni (3000 223 ppm) to investigate their anti-heavy metals contamination performance differences. How was the concentration of vanadium and nickel adsorbed on the catalysts determined? What method was used to incorporate the vanadium and nickel into the kaolin structure? This information is not found in the methodology section; such as how many catalysts were synthesized? Were the reaction measurements sufficient to determine the standard deviation or experimental error, or was it just one measurement?
Author Response
Reviewer 1:
The manuscript is interesting, especially the modification of kaolin by Si exchange in its structure and the application of the FCC reaction, due to the multiple diffusion and reaction factors that these generate in interaction with catalysts. However, it is necessary to consult a greater amount of literature, more updated, as the most recent is from 2019, two years from the current date. There is a lot of literature related to FCC reactions that could explain some of the phenomena observed in his study.
It is necessary to refine the XRD results to reduce the noise because it is confused with the crystallinity that is mentioned in the angles of 13.5o, 38.0o, 49.3o and 65.4o.
Comments:
- In section 2.2. Synthesis of the Si-PB@kaolin compound; line 69 mentions: After that, the Na2SiO3 and NaAlO2 solutions were added simultaneously into the above suspension until the pH of the system approached about 8.5, however, it is not specified at what relation weight or mol, the Na2SiO3, and NaAlO2 were added to kaolin. And how the pH of 8.5 is reached or what reagent is added to reach it? You need to complete this section with more information to clarify.
Reply: Thank you for your comments. We have given specific synthesis information in corrected manuscript.
- In line 99 it is mentioned that the catalyst/oil catalytic cracking weight ratio being 5, that is, for each portion of catalyst, 1/5 of oil was added? How is this relationship explained or why was this relationship used?
Reply: “catalyst/oil weight ratio being 5” suggested that the catalyst and feed oil in ACE evaluation unit were added at the ratio of “1 g catalyst for 5 g feed oil”, which is referenced to industrial FCC unit.
- Line 113, mentioned: For calcinated kaolin and Si-PB@kaolin composite samples, the characteristic diffraction peaks of kaolin disappeared leaving a wide diffraction peak in the range of 15-30 due to the destruction structure caused by calcination. But what happens during the calcination process until 800 °C, How is it transformed into semi amorphous material?
Reply: It had been reported that (Appl Clay Sci,2001,19:89 – 94), when the calcination temperature exceeded 800 °C, the crystalline structure of kaolin began to collapse because of the breaking of Al-O bond, and thus transformed to amorphous phase (meta-kaolin).
- Line 118 mentions that several new wide diffraction peaks appeared at around 13.5 o, 38.0 o, 49.3 o and 65.4 o respectively for Si-PB@kaolin composite, which could be attributed to the characteristic diffractions of Si-PB and suggested the successful construction of Si-PB secondary units into the structure of kaolin. But what will be the crystalline system that it presents? will it change from monoclinic to some other system or network by structural reorganization?
Reply: Because these new diffraction peaks are in good with agreement with the diffraction characteristics of pseudo-boehmite phase. Therefore, we believe that these new diffraction peaks could be attributed to Si-PB.
- Line 138 mentions that: On the contrary, due to the introduction of secondary Si-PB units, the surface area and pore volume for the Si-PB@kaolin compound obviously increased to 106 m2/g and 0.28 cm3 /g respectively, which were much higher than that of conventional kaolin and calcined kaolin. But do you mean that these secondary units are the ones that contribute to the surface area? How to know if in DRX it was commented that these are identified as independent structures? Can these units be found inside kaolin as illustrated in Scheme 1? can you comment something about this?
Reply: Because the surface area and pore volume for conventional kaolin and calcined kaolin are comparatively low. As a result, the increasing of surface area and pore volume for Si-PB@kaolin could be mainly attributed to the introduction of Si-PB secondary units. It can be seen from XRD results (Figure 1) that there were not XRD patterns of Si-PB secondary units for conversional kaolin and calcinated kaolin, which could suggest that Si-PB secondary units are not existed in conversional kaolin and calcinated kaolin. As a result, Si-PB secondary units could be identified as independent structures.
- Figure 4. Py-FTIR spectra for kaolin, AT-kaolin and Si-PB@kaolin, the signal between 1400 and 1500 cm-1, to which species does it correspond or what could it indicate?
Reply: Based on the report (Micropor Mesopor Mater, 2005, 82(1/2): 99-104) that, this signal could be attribute to the combined adsorption of B+L acid sites. However, the quantifies for B and L acid sites are usually made by 1540 and 1450 cm-1 respectively.
- Line 184 indicates that the value of Cat-2 was much bigger than that of Cat-1, suggesting that Si-PB@kaolin could obviously enhance the acid sites accessibility of prepared FCC catalyst in comparison with conventional kaolin matrix material. Can you explain, why is it acidity changed? what bonds in its union will be present or are the SiPB blocks that cause said the increase, and how are they organized or structurally united?
Reply: Because of the much bigger pore volume of Si-PB@kaolin in compared with conventional kaolin, as matrix material, the Si-PB@kaolin would obviously promote the diffusion property of the prepared FCC catalyst by improving the pore structure of FCC catalyst, and thus enhance the acid sites accessibility of the prepared FCC catalyst.
- From the Ca2 sample. It would be interesting to show a TEM to observe its structural organization.
Reply: Thank you for your advice. However, the operator of TEM told us that our FCC catalyst was not very suitable for TEM analysis because of its composition, particle size and shape etc.
- The prepared FCC catalysts were also contaminated by V (5000 ppm) and Ni (3000 223 ppm) to investigate their anti-heavy metals contamination performance differences. How was the concentration of vanadium and nickel adsorbed on the catalysts determined? What method was used to incorporate the vanadium and nickel into the kaolin structure? This information is not found in the methodology section; such as how many catalysts were synthesized? Were the reaction measurements sufficient to determine the standard deviation or experimental error, or was it just one measurement?
Reply: The V and Ni contaminated species were incorporated into FCC catalyst by Impregnation method, and their concentrations ware determined by XRF. The reaction measurement used in this work is multichannel, and can load 6 catalysts samples in a batch.

Reviewer 2 Report
This work aims to synthesize a novel matrix material for FCC catalyst based on Si-modified pseudo-boehmite@kaolin composite, in order to overcome some drawbacks of fluid catalytic cracking process, such as hydrothermal stability and physical and acid properties.
The work is worth of publication after reviewing some of the following aspects and recommendations:
1- I strongly recommend to thoroughly revise English and some typos and vocabulary (for example, “extracting”, “calcinated” (it should be calcined)”, “percentage points” (maybe directly %), etc.
- Please also avoid some repetitions (for example line 60 and 63, “all chemicals were used as received…”)
- Materials and Methods. I am missing some information about calcinations procedure, which is crucial for both matrix and catalysts.
- Characterization and evaluations. Please refer to prior works or shortly extend the procedure followed for catalyst and matrix characterization.
Also complete the operating conditions: space time, residence time, the number of repetitions for each kinetic run, etc.
A description of the reaction equipment will also be appreciated.
For heavy oil characterization, please provide who supplied this feed, refer to ASTM standards for the characterization (density, etc.). Elemental analysis (C,H, N, S) is missing.
Did the authors carried out Distillation simulation analysis for both feed and product analysis? How was the sulphur content determined?
Information about product analysis is also missing: type of chromatographs used, sulfur content of products (PFPD detector?), definition of lumps of products (dry gas, LPG, etc.) including C number for each lumped product.
Please also provide information about how product yield is calculated.
- All figures should include Y axis label, eventhough arbitraty units are plotted (intensity, adsorbed volume,….)
- For N2 adsorption-desorpiton analysis please refer to new classification of IUPAC for isotherms and replace the literature citation.
- I recommend to show Table 2 and 3 as a sole table for physical and acid properties characterization. Same for Table 4 and missing table of acid properties for catalysts.
- Please add more information on FCC catalysts in Table 4: Vmicro, Vmeso, SBET, Sext, etc.
- Define MAT abbreviation
- Include the information provided in Figure 6 in Table 4 and complete with FTIR and TPD analysis information (total acidity, B/L ratio, etc.)
- About kinetic results, I suggest to show Table 5 and Table 6 results as figures instead of tables. I also recommend to improve table or figure caption.
- What is “Total Liquid” in Table 5. Please explain it on experimental or throughout the text.
- Did the authors carry out PONA analysis or similar to identifiy the molecular families within gasoline (aromatic,s paraffins, olefins, naphthenes? This information is relevant for FCC process.
- The procedure to calculate the coke content should be described in the manuscript, and I strongly recommend to show the TPO profiles for Cat-1 and Cat-2 to analysis coke deactivation.
- The results shown should be discussed extensively and compared with prior works in the literature. The studies from Prof. J.M. Arandes and Prof. J. Bilbao, from the UPV/EHU could be a good reference point or also the studies from Prof. U. Sedrán in Argentina. This way, updated bibliography would be added into the manuscript. New references from 2020-2022 should be included as FCC is a very important catalytic process widely investigated.
- Conclusions should be rewritten in coherence to the benefits of this new matrix synthesized.
Regards
Author Response
Reviewer 2:
This work aims to synthesize a novel matrix material for FCC catalyst based on Si-modified pseudo-boehmite@kaolin composite, in order to overcome some drawbacks of fluid catalytic cracking process, such as hydrothermal stability and physical and acid properties.
The work is worth of publication after reviewing some of the following aspects and recommendations:
1- I strongly recommend to thoroughly revise English and some typos and vocabulary (for example, “extracting”, “calcinated” (it should be calcined)”, “percentage points” (maybe directly %), etc.
Reply: Thanks for your advice. We have made corresponding corrections in corrected manuscript.
- Please also avoid some repetitions (for example line 60 and 63, “all chemicals were used as received…”)
Reply: Thanks for your advice. We have deleted the repetitions of “All chemicals in this study were used as received” in corrected manuscript.
- Materials and Methods. I am missing some information about calcinations procedure, which is crucial for both matrix and catalysts.
Reply: Thank you for your comments. We have added more calcinations information in corrected manuscript.
- Characterization and evaluations. Please refer to prior works or shortly extend the procedure followed for catalyst and matrix characterization.
Also complete the operating conditions: space time, residence time, the number of repetitions for each kinetic run, etc.
A description of the reaction equipment will also be appreciated.
For heavy oil characterization, please provide who supplied this feed, refer to ASTM standards for the characterization (density, etc.). Elemental analysis (C,H, N, S) is missing.
Did the authors carried out Distillation simulation analysis for both feed and product analysis? How was the sulphur content determined?
Information about product analysis is also missing: type of chromatographs used, sulfur content of products (PFPD detector?), definition of lumps of products (dry gas, LPG, etc.) including C number for each lumped product.
Please also provide information about how product yield is calculated.
Reply: Thank you for your comments. All needed information is added in corrected manuscript.
- All figures should include Y axis label, eventhough arbitraty units are plotted (intensity, adsorbed volume,….)
Reply: Thank you for your comments. We have made corrections in corrected manuscript.
- For N2 adsorption-desorpiton analysis please refer to new classification of IUPAC for isotherms and replace the literature citation.
Reply: Thank you for your comments. We have cited new literature of IUPAC in corrected manuscript.
- I recommend to show Table 2 and 3 as a sole table for physical and acid properties characterization. Same for Table 4 and missing table of acid properties for catalysts.
Reply: Thank you for your comments. We have made corrections in corrected manuscript.
- Please add more information on FCC catalysts in Table 4: Vmicro, Vmeso, SBET, Sext, etc.
Reply: Thank you for your comments. Corresponding information have been provided in Table 3 in corrected manuscript.
- Define MAT abbreviation
Reply: Thank you for your comments. The MAT had been defined in section 2.4.
- Include the information provided in Figure 6 in Table 4 and complete with FTIR and TPD analysis information (total acidity, B/L ratio, etc.)
Reply: The total acidity and B/L have been given in Table 3 of corrected manuscript.
- About kinetic results, I suggest to show Table 5 and Table 6 results as figures instead of tables. I also recommend to improve table or figure caption.
Reply: Thank you for your comments. We have changed Table 5 and Table 6 into corresponding figures (Figure 8 and Figure 9) in corrected manuscript.
- What is “Total Liquid” in Table 5. Please explain it on experimental or throughout the text.
Reply: The explanation of “Total Liquid” has been give in Table 4 in corrected manuscript.
- Did the authors carry out PONA analysis or similar to identifiy the molecular families within gasoline (aromatic,s paraffins, olefins, naphthenes? This information is relevant for FCC process.
Reply: The PONA analysis of gasoline has been added in Table 6 in corrected manuscript.
- The procedure to calculate the coke content should be described in the manuscript, and I strongly recommend to show the TPO profiles for Cat-1 and Cat-2 to analysis coke deactivation.
Reply: The calculation method for coke has been added in section 2.4 in corrected manuscript.
- The results shown should be discussed extensively and compared with prior works in the literature. The studies from Prof. J.M. Arandes and Prof. J. Bilbao, from the UPV/EHU could be a good reference point or also the studies from Prof. U. Sedrán in Argentina. This way, updated bibliography would be added into the manuscript. New references from 2020-2022 should be included as FCC is a very important catalytic process widely investigated.
Reply: Thank you for your comments. We have cited and discussed corresponding literatures in introduction of corrected manuscript.
- Conclusions should be rewritten in coherence to the benefits of this new matrix synthesized.
Reply: Thanks for your advice. We have added the corresponding discussion in corrected manuscript.

Reviewer 3 Report
Title: Synthesis of Si-modified pseudo-boehmite@kaolin composite…
Manuscript ID: materials-1612845
Authors: Yuan et al.
Dear Authors,
Thank you for the opportunity to read your article. I found the topic is interesting and fundamental. Generally speaking, there are some results presented in order to capture some trends but the methods and results need more clear explanation while the results need more clear and detail discussion with fair point of view. I suggest that this article will be revised extensively before its re-submission for another review process if applicable. As a conclusion, I recommend its major revision at this state.
I hope my comments are helpful.
Good luck,
A reviewer
Major concerns:
“Article title”
-Is it common to use “@” in the article title?
“Abstract”
-Line 19: “After V…”->Please consider mentioning what is “V” in this statement briefly.
“Keywords”
->Please consider providing keywords that are not used in the article title.
“1. Introduction”
-Lines 46-48: “…it has been reported that Si-modification could obviously improve…”->As Si-modification has been already reported, please consider stating (a) research gap(s) you tried to address and (b) the originality of this work in this section.
“2. Materials and methods”
“2.4. Characterizations and evaluations”
-In this section, please consider providing more details about the measurements. For example, please consider providing more detail information about your SEM imaging, including the way you deposited your sample(a) in an SEM chamber, detector type (SE? BSE?), accelerating voltage, working distance. Those information would be helpful for future researchers. This comment also applied to all the characterization methods introduced in this section.
doi.org/10.1016/j.actamat.2005.12.014
doi:10.3390/electronics8101202
“3. Results”-> “3. Results and discussion”?
In general, the explanation and discussion of the results are rather limited except the results shown in Figure 4. Please consider discussing your results more. Also, there is no obvious overarching discussion among different results obtained by using different characterization methods. You may find my detail comments below for your help.
“3.1.Characterization results”
-Figure 1 (and other figures): Please name the y-axis value and provide its unit.
-Lines 149-150: “…the NH3-deposition peak area…was much higher, indicating that there were much more surface acid sites on Si-PB@kaolin composite.”->Please consider citing literature values and compare them with your values. Also, please try to link this result with your results shown in Figure 4 if appropriate.
“3.2. Characterizations for prepared FCC catalysts”
-Table 4: Please consider defining “MAT%” clearly in this section.
-Comparing the properties between “Cat-1” and “Cat-2”, there are some differences but not so significant although the results shown in Tables 2 and 3 indicate the significant differences between them. How do you explain this difference in the significance among the different results?
“3.3. Evaluation results for heavy oil catalytic cracking”
-Table 6: There are some results (i.e. Diesel, Bottom) showing that “Cat-1” is better than “Cat-2”. How do you explain it? Is it against your expectation? You may compare your results with literature results in this section.
“4. Conclusions”
-You may state future perspectives in Conclusions.
Minor concerns:
-Please consider polishing English more. You may use some of my comments above for this purpose.
Author Response
Reviewer 3:
- Dear Authors,
- Thank you for the opportunity to read your article. I found the topic is interesting and fundamental. Generally speaking, there are some results presented in order to capture some trends but the methods and results need more clear explanation while the results need more clear and detail discussion with fair point of view. I suggest that this article will be revised extensively before its re-submission for another review process if applicable. As a conclusion, I recommend its major revision at this state.
- I hope my comments are helpful.
- Good luck,
- A reviewer
- Major concerns:
- “Article title”
- -Is it common to use “@” in the article title?
- Reply: Yes. In some accepted papers, the composite between different substances can be used by @, such as ACS Appl. Nano Mater. 2021, 4, 6306.
- “Abstract”
- -Line 19: “After V…”->Please consider mentioning what is “V” in this statement briefly.
- Reply: Thank you for your comments. We have made correction in corrected manuscript.
- “Keywords”
- ->Please consider providing keywords that are not used in the article title.
- Reply: Thank you for your comments. We have made supplement in corrected manuscript.
- “1. Introduction”
- -Lines 46-48: “…it has been reported that Si-modification could obviously improve…”->As Si-modification has been already reported, please consider stating (a) research gap(s) you tried to address and (b) the originality of this work in this section.
Reply: Currently, kaolin is the main matrix material for FCC catalyst, and always suffers from the drawbacks such as small surface area, low pore volume and deficient acid sites. Therefore, to overcome the above drawbacks of conventional kaolin, the Si-modified pseudo boehmite unit that with big surface area, high pore volume and sufficient acid sites was constructed in the structure of conventional kaolin in situ.
This content has been supplied in corrected manuscript.
- “2. Materials and methods”
- “2.4. Characterizations and evaluations”
- -In this section, please consider providing more details about the measurements. For example, please consider providing more detail information about your SEM imaging, including the way you deposited your sample(a) in an SEM chamber, detector type (SE? BSE?), accelerating voltage, working distance. Those information would be helpful for future researchers. This comment also applied to all the characterization methods introduced in this section.
Reply: More details of measurement have been supplied in corrected manuscript.
- org/10.1016/j.actamat.2005.12.014
- doi:10.3390/electronics8101202
- “3. Results”-> “3. Results and discussion”?
- In general, the explanation and discussion of the results are rather limited except the results shown in Figure 4. Please consider discussing your results more. Also, there is no obvious overarching discussion among different results obtained by using different characterization methods. You may find my detail comments below for your help.
Reply: Thank you for your comments. We have added corresponding explanation and discussion in corrected manuscript.
- “3.1.Characterization results”
- -Figure 1 (and other figures): Please name the y-axis value and provide its unit.
Reply: Thank you for your comments. We have made corrections in corrected manuscript.
- -Lines 149-150: “…the NH3-deposition peak area…was much higher, indicating that there were much more surface acid sites on Si-PB@kaolin composite.”->Please consider citing literature values and compare them with your values. Also, please try to link this result with your results shown in Figure 4 if appropriate.
Reply: Thank you for your good advice. We have cited the literature in corrected manuscript.
- “3.2. Characterizations for prepared FCC catalysts”
- -Table 4: Please consider defining “MAT%” clearly in this section.
Reply: It is stated that in the section of “2.4. Characterizations and evaluations” that “MAT” is micro activity test and used to evaluate the cracking activity of FCC catalyst.
- -Comparing the properties between “Cat-1” and “Cat-2”, there are some differences but not so significant although the results shown in Tables 2 and 3 indicate the significant differences between them. How do you explain this difference in the significance among the different results?
Reply: Tables 2 and 3 only display the differences between matrix materials. However, FCC catalysts (Cat-1 and Cat-2) are mixtures of multi-component including zeolites, binder and matrix etc, which would make the differences between Cat-1 and Cat-2 not be so significant.
- “3.3. Evaluation results for heavy oil catalytic cracking”
- -Table 6: There are some results (i.e. Diesel, Bottom) showing that “Cat-1” is better than “Cat-2”. How do you explain it? Is it against your expectation? You may compare your results with literature results in this section.
Reply: For heavy oil catalytic cracking reaction, the diesel and bottom are consider as heavy products. Higher yields of diesel and bottom mean lower heavy oil cracking ability for FCC catalyst. Therefore, in this work, the higher yields of diesel and bottom for Cat-1 in compared with Cat-2 would suggest that the our Si-PB@kaolin composite has effectively increased the heavy oil cracking ability of the prepared FCC catalyst (Cat-2), which is in good agreement with our expectation.
- “4. Conclusions”
- -You may state future perspectives in Conclusions.
Reply: We have given the statement about future perspectives in corrected manuscript.
- Minor concerns:
- -Please consider polishing English more. You may use some of my comments above for this purpose.
Reply: Thank you very much for your help. In the main text, we have strengthened the English writing and added relevant discussion.

Round 2
Reviewer 2 Report
Authors have thoroughly revised the manuscript following the reviewer's suggestions.
Reviewer 3 Report
Dear Authors,
As all the concerns were addressed, I suggest the journal accept this article for its publication.
Best regards,
A reviewer